# Design and Assist-as-Needed Control of Flexible Elbow Exoskeleton Actuated by Nonlinear Series Elastic Cable Driven Mechanism

Bingshan Hu [1,2,*], Fuchao Zhang [1], Hongrun Lu [1], Huaiwu Zou [3], Jiantao Yang [1,2] and Hongliu Yu [1,2]

1   Institute of Rehabilitation Engineering & Technology, University of Shanghai for Science and Technology, Shanghai 200093, China; 202562472@st.usst.edu.cn (F.Z.); luhongrun_lewis@163.com (H.L.); jty@usst.edu.cn (J.Y.); yhl98@hotmail.com (H.Y.)
2   Shanghai Engineering Research Center of Assistive Devices, Shanghai 200093, China
3   Shanghai Aerospace System Engineering Research Institute, Shanghai 201109, China; shanshuijun@163.com
*   Correspondence: hubingshan@usst.edu.cn

**Abstract:** Exoskeletons can assist the daily life activities of the elderly with weakened muscle strength, but traditional rigid exoskeletons bring parasitic torque to the human joints and easily disturbs the natural movement of the wearer's upper limbs. Flexible exoskeletons have more natural human-machine interaction, lower weight and cost, and have great application potential. Applying assist force according to the patient's needs can give full play to the wearer's remaining muscle strength, which is more conducive to muscle strength training and motor function recovery. In this paper, a design scheme of an elbow exoskeleton driven by flexible antagonistic cable actuators is proposed. The cable actuator is driven by a nonlinear series elastic mechanism, in which the elastic elements simulate the passive elastic properties of human skeletal muscle. Based on an improved elbow musculoskeletal model, the assist torque of exoskeleton is predicted. An assist-as-needed (AAN) control algorithm is proposed for the exoskeleton and experiments are carried out. The experimental results on the experimental platform show that the root mean square error between the predicted assist torque and the actual assist torque is 0.00226 Nm. The wearing experimental results also show that the AAN control method designed in this paper can reduce the activation of biceps brachii effectively when the exoskeleton assist level increases.

**Keywords:** assist-as-needed control; exoskeleton; flexible; musculoskeletal model; series elastic actuator





## 1. Introduction

The degeneration of musculoskeletal system caused by aging affects the daily living activities of some elderly people seriously. Exoskeletons can provide power assistance for the wearer, but traditional rigid exoskeletons are easy to impose additional motion constraints on the wearer's joints, resulting in parasitic torque [1]. At present, passive degrees of freedom, self-aligning mechanisms, or remote rotation center are used to solve the above problems, but the size and mass are increased [2]. The flexible exoskeleton is driven by flexible parts, which eliminates the need of precise alignment between robot and biological joint axis and has better ergonomics and cost advantages [3].

The flexible assist exoskeleton includes pneumatic drive type [4–6], functional material drive type [7,8], cable drive type, and so on [9]. At present, most flexible exoskeletons of upper limbs are driven by a Bowden cable. The driving principle of a Bowden cable is to transmit the motor driving torque to the joints of upper limbs through cables, winding wheels, and pulleys [10]. The cable has an outer sheath, which can effectively reduce friction and prevent interference between the cable and other parts. Early flexible exoskeletons of upper limbs were usually driven redundantly by cable actuators [11], and the configuration design was optimized to meet the motion freedom and workspace of human upper limb joints so that the exoskeletons have the motion characteristics similar to human arms in the

macro motion level [12]. One of the development trends of flexible upper limb exoskeleton is to arrange the cable actuator along the tension line of the upper limb primary motor muscle to assist the corresponding muscles directly, referring to the human musculoskeletal system. The human skeleton is used as the support structure of the flexible exoskeleton, so the human-machine hybrid structure conforms to the natural flexibility of the human body with a rigid base [13]. Researchers have found that wearing this exoskeleton can reduce muscle activation by 58.17% under the same load [14].

In terms of assist force control, rehabilitation medicine research shows that if we can make full use of the residual muscle strength of patients, it is more conducive to muscle strength training and motor function recovery than if the assist force is completely provided by exoskeleton [15]. Therefore, researchers put forward the strategy of assist-as-needed (AAN) control (that is, the assist force provided by robot is determined and controlled according to the physiological condition of the wearer and the specific situation of the task) [16]. The main difficulty of this method is to accurately predict the gap between the force needed to complete the task and the wearer's own muscle strength.

According to different assist force prediction methods, AAN control can be divided into assist force predictive control based on exoskeleton force feedback and model free assist force predictive control based on both surface electromyography (sEMG) signal and on human musculoskeletal model [17]. The force feedback control calculates the human-robot interaction force/torque by establishing an accurate dynamic model and measuring the kinematic and dynamic information using various sensors installed on the robot, such as force/torque, position, speed, etc. Then, a human-robot interaction control strategy can be designed [18,19]. However, this method requires high accuracy of the dynamic model, and the feedback force/torque is static lag information, which can be detected only when the wearer initiates an action to trigger the response of the exoskeleton. In addition, this method is generally used on the rigid exoskeleton.

The model-free approach uses machine learning for approximating a numerical function that maps between sEMGs to joint torques. The mapping function can be approximated by the combination of linear to polynomial or exponential to artificial neural network, without specific function basis [20–22]. All of the intermediate functional relationships between experimental variables observed in the model free method are not explicitly modeled, but are included in the macro transfer function. Thus, the main problem is that the functional relationships learned in specific conditions or regions may not be extended to new conditions [23]. In addition, this method does not allow adjusting the assist level of the robot, resulting in unnatural human-robot interaction [24].

With the development of biomechanics, researchers have established different kinds of human upper limb musculoskeletal models, combined with joint kinematics and sEMG signals reflecting the muscle activation degree which can realize the prediction of human upper limb muscle force and joint torque [25]. Using the musculoskeletal model to predict the assist force and map it to the control command of the exoskeleton is a more closely coupled and natural method of human-robot interaction [26]. AAN control based on musculoskeletal model needs to collect a large number of motion parameters and human physiological parameters to build a complex musculoskeletal model of upper limb, and the process of muscle strength estimation is complex [27,28]. In order to meet the needs of actual control, it is necessary to simplify and optimize the multi degree of freedom upper limb musculoskeletal model reasonably. For example, the common double musculoskeletal model is often used in elbow joint [29], and the common musculoskeletal model parameter optimization methods include linear optimization, nonlinear optimization based on genetic algorithm, nonlinear least square optimization, and other methods [30]. After the joint assist torque is predicted, it needs to be mapped to the cable actuator and controlled. Here, a force sensor can be placed to measure the force on the cable transmission path. However, the rigid force sensor increases the cost and violates the principle of flexible wearability [31]. Using the principle of series elastic actuating, the complex force control can be transformed

into position feedback control. At present, some cable driven exoskeletons have used this method [32–34].

In this paper, a flexible elbow exoskeleton is designed, which is driven by two nonlinear cable series elastic actuators (SEA) antagonistically. The innovation of this paper is that the SEA simulates the passive elastic properties of human skeletal muscle, and an improved musculoskeletal model was used to predict the muscle torque of elbow joint, which included six muscles and was more consistent with the anatomical characteristics of elbow joint. Finally, the force of the nonlinear SEA is controlled without force sensor to assist the elbow flexion and extension muscles directly.

The rest of this paper is organized as follows: Section 2 introduces the structural design of the exoskeleton and the driving principle of the nonlinear SEA. In Section 3, the AAN control method is introduced in detail, including assist torque prediction based on improved elbow musculoskeletal model, assist torque decomposition and nonlinear cable SEA force control. Section 4 introduces the experimental scheme and results. Finally, conclusions and future research ideas are given.

## 2. Design of the Cable Driven Flexible Exoskeleton

### 2.1. Overall Structural Design

As shown in Figure 1a, the elbow exoskeleton designed in this paper is composed of an actuating control box and a wearable part. The actuating control box (Figure 1b) mainly includes two nonlinear cable SEAs, two motor controllers, a central controller, and a battery. The force transmission path of the cable SEA is arranged along the tension line of the upper arm flexion and extension muscles. The torque of the two SEAs acting on the human elbow joint corresponds to the resultant torque of the flexion and extension muscles, respectively. The starting and ending points of the driving cable are determined based on the human anatomical model and are arranged near the starting and ending points of the assisted muscles. The wearable part includes wearable fabric, cable, sheath, anchor points and inertial sensor (Figure 1a). The anchor points and sheath are fixed on the surface of wearable fabric to guide the cable to move along the designed driving path. The inertial sensor is to measure the elbow angle.

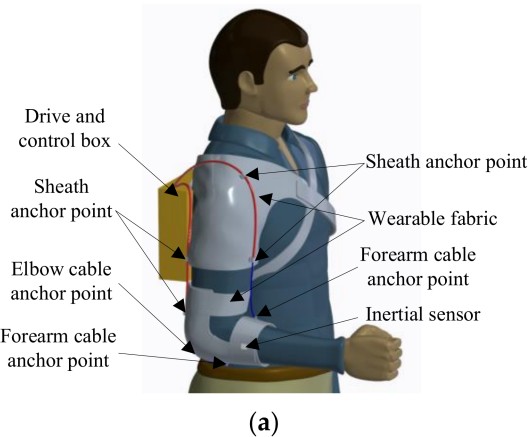

(**a**)

**Figure 1.** *Cont.*

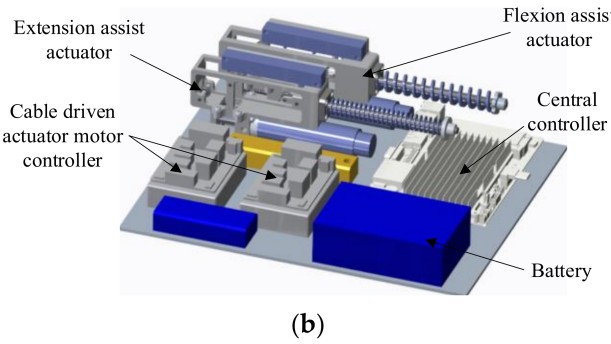

(**b**)

**Figure 1.** Schematic diagram of cable driven flexible exoskeleton. (**a**) Overall structure diagram. (**b**) Schematic diagram of actuating control box.

### 2.2. Principle of the Nonlinear Cable Sea Mechanism

The cable drive mechanism in Figure 1 adopts the principle of series elastic actuating, which realizes force control without adding a force sensor. As shown in Figure 2a, the nonlinear series elastic cable actuator includes a motor, a planetary reducer, a winding wheel, a cable, a linear guide rod, a linear bearing, a linear compression spring, and three pulleys. The winding wheel is directly connected to the reducer output rotating shaft, and the motor's rotation torque is transformed into the cable's force through the winding wheel (Figure 2b). The linear spring passes through the linear guide rod. One end of the linear guide rod passes through the linear bearing and connects with the pulley twp, and they connect with a linear displacement sensor. The motor in the actuator is CBL2453-2423f motor from TECHSERVO company, and it is equipped with a gear reducer. The rated output torque of the motor is 27.28 mNm, and the reduction ratio of the reducer is 100. The KSF50 of Miran company is selected as the linear displacement sensor, which is based on the principle of potentiometer. Its stroke is 50 mm, and its linear accuracy reaches 0.1%.

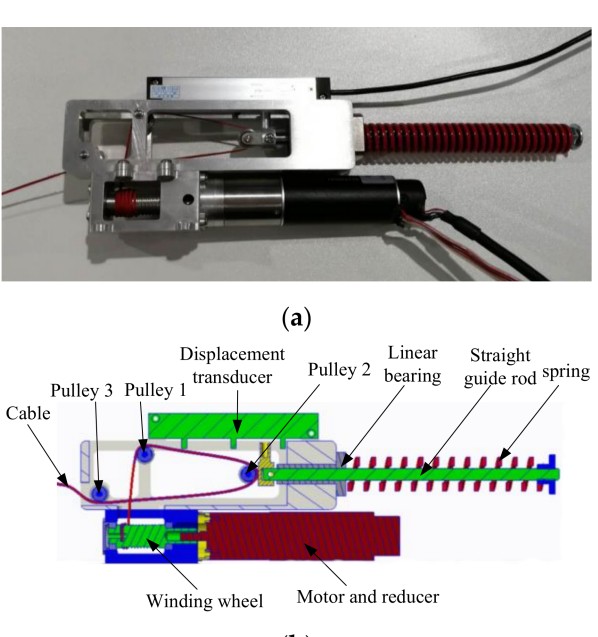

**Figure 2.** Nonlinear series elastic cable drive mechanism. (**a**) Physical drawing. (**b**) Sectional drawing.

According to the classical Hill muscle model (Appendix A), muscle contraction force includes both an active contraction force and passive contraction force. The simulation of the active contraction force can be realized by the active control of the motor, so that the relationship between the active contraction force and displacement of the cable driving mechanism conforms to the Hill muscle model. The passive contraction force increased

exponentially with the increase of muscle fiber contraction length [35]. According to the characteristic curve of muscle passive contraction force in Figure 3, when the muscle length is less than the rest length (the elbow flexion angle is about 60°), there is no passive force. The cable actuator is designed so that there is no output displacement when the elbow flexion angle is about 60°. When the elbow flexion angle is greater than 60°, due to the unidirectional force transmission characteristics of the cable actuator, there is no output force on the cable. When the elbow flexion angle is less than 60°, the nonlinear stiffness characteristics of the cable actuator are fitted with the force-displacement characteristics of the muscle stretch by optimizing the parameters in the actuator.

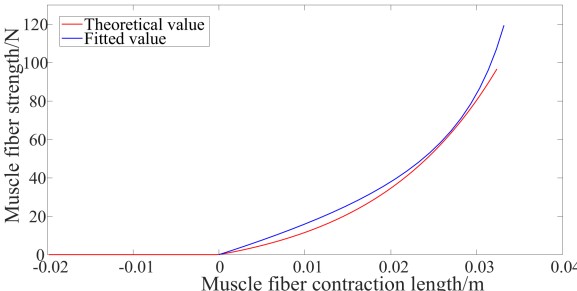

**Figure 3.** Passive contraction force of skeletal muscle (Long head of biceps brachii) and its fitting. Muscle fiber contraction length = 0 means that the muscle is at rest (that is, the elbow flexion is about 60 degrees).

In this paper, the elastic elements in the cable SEA mechanism are designed as nonlinear elements. The nonlinear spring needs to be customized according to particular needs, and its nonlinearity is difficult to fully meet the actual needs. In some literatures, the triangular pulley block and linear spring are used to form the nonlinear spring mechanism [35], and this design method is also used in this paper. As shown in Figure 2b, the cable driving mechanism in this paper uses three pulleys to form a pulley block, which changes the transmission path of the cable force output into a triangle, and the pulley two is directly connected with the linear spring. When the cable pulls the linear guide rod to move, the spring compression changes and the output stiffness of the cable SEA is adjusted.

As shown in Figure 4, let the centers of the three pulleys be points $A_C$, $B_C$ and $C_C$, respectively, and ignore the friction here. Assuming that the cable force at the pulley $A_C$ and $C_C$ is $F_C$, and the spring force at the pulley $C_C$ is $Fe$. Let the initial vertical distance between $A_C B_C$ be $b$, the vertical distance between $A_C C_C$ be $c$, the horizontal distance between $A_C B_C$ and the horizontal distance between $A_C C_C$ be $a$, and the angle between the connecting line of $B_C C_C$ and the vertical line be $\alpha$. The angle between $A_C B_C$ line and vertical line is $\beta$. After spring compression, the real-time vertical distance between $A_C B_C$ points is $\xi$, and the value range of $\xi$ is $[0, b]$. The elastic coefficient of the spring is $k$. According to the force analysis of the cable SEA in Figure 4, the cable force $F_C$ can then be deduced as follows:

$$F_c = \frac{k(b-\xi)\sqrt{(a^2+\xi^2)[a^2+(\xi+c)^2]}}{\xi(\sqrt{a^2+(\xi+c)^2}+\sqrt{a^2+\xi^2})+c\cdot\sqrt{a^2+\xi^2}} \tag{1}$$

It can be seen from Equation (1) that the parameters affecting $F_C$ include $k$, $a$, $b$, $c$, and $\xi$. Among them, $\xi$ is the variation, which reflects the displacement of the cable actuator and can be measured by the linear displacement sensor. The parameters $k$, $a$, $b$, and $c$ determine the passive elastic properties of the mechanism. After optimization, $k$, $a$, $b$, and $c$ are taken as 4250 N/m, 0.01 m, 0.02 m and 0.026 m, respectively. The passive contraction force-displacement curve of cable SEA mechanism is shown in the blue line in Figure 3, which can simulate the passive contraction force-displacement relationship of muscle (red line in Figure 3) very well.

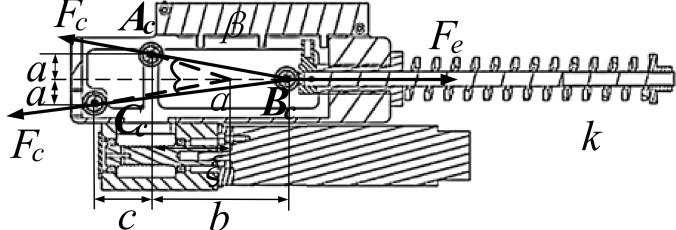

**Figure 4.** Force analysis of cable SEA mechanism.

## 3. Modeling of Shape Memory Alloy Spring Actuator

As shown in Figure 5, the assist-as-needed force control system of the flexible exoskeleton in this paper mainly includes three modules: assist torque prediction based on elbow musculoskeletal model, assist torque decomposition and force control of nonlinear cable SEA, and the assist-as-needed control principle can be expressed by Equation (2).

$$T_{assist} = T_G - T_{muscle} = \begin{bmatrix} r_B & r_T \end{bmatrix} \begin{bmatrix} F_B & F_T \end{bmatrix}^T \tag{2}$$

In the assist torque prediction module, the position command of elbow joint $\theta_{elbowref}$ should be the input. According to the weight of the arm and the weight of the load on the hand, the load torque $T_G$ of the elbow joint was calculated. Here, the inertial force, Coriolis force, and centrifugal force during arm movement are ignored, and it is assumed that there is no contact force between the end of the arm and the outside environment. The arm load is mainly the weight of the forearm itself and the gravity at the end of the hand. Then, according to the real-time sEMG of the main muscles of elbow joint, and elbow joint angle $\theta_{elbow}$, the joint muscle resultant moment $T_{muscle}$ was predicted according to the elbow musculoskeletal model. Finally, $T_{assist}$ was predicted according to the difference between joint load torque and elbow muscle torque. If $T_{muscle}$ is less than $T_G$, it means assistance is needed.

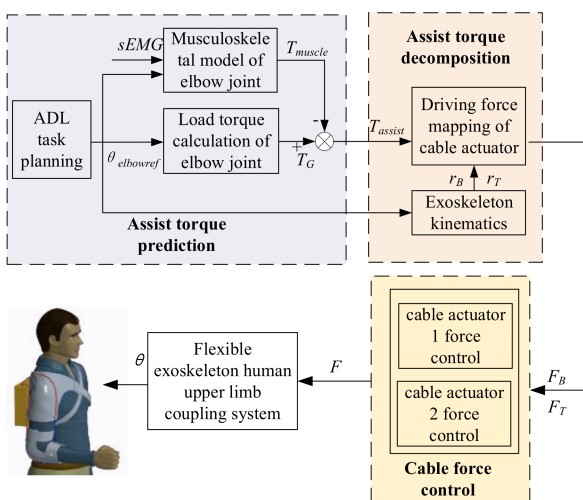

**Figure 5.** Assist-as-needed control system block diagram.

In the assist torque decomposition module, the real time flexion and extension cable force arm $r_B$ and $r_T$ can be calculated according to the kinematics of flexible exoskeleton. Then using Equation (2), the assist force command $F_B$ and $F_T$ of the two-cable actuator on the flexible exoskeleton are obtained.

Finally, the assist force of the cable actuator is controlled, which will be introduced in detail later.

### 3.1. Prediction of Exoskeleton Assist Torque

As shown in Figure 6, in order to predict the muscle torque of elbow joint accurately, this paper improves the double muscle elbow joint musculoskeletal model established by Borzelli [29], and an improved musculoskeletal model of elbow joint is established. The model includes six muscles (the biceps brachii muscle strength is divided into two muscles, the triceps brachii muscle is divided into three muscles, and the brachii muscle is added as flexor), so the number of muscles is more complete. At the same time, considering the existence of humeral trochlea, the geometric model is more in line with the elbow physiological structure.

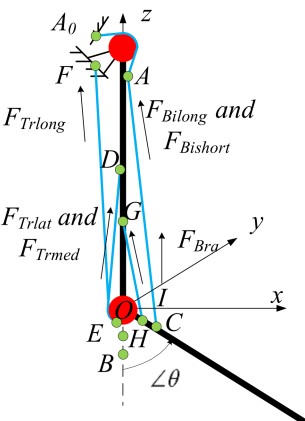

**Figure 6.** Schematic diagram of improved elbow joint musculoskeletal model.

In Figure 6, the origin $O$ is the rotation center of elbow joint, and the coordinate system $O$-$xyz$ is established. $A_0$ is the starting point of long head and short head of biceps brachii, $B$ is the insertion point of biceps brachii, and $C$ is the position after rotation of $B$. Since the biceps brachii is a double joint muscle, in order to eliminate the influence of shoulder joint movement on the length of biceps brachii, we assume that the shoulder joint is fixed, an additional point $A$ is set as the starting point of biceps brachii, and the length from $A_0$ to $A$ is regarded as constant when the elbow joint moves. $A$ is used to calculate the muscle length change. $D$ is the starting point of medial head and lateral head of triceps brachii. $F$ is the starting point of long head of triceps brachii. $E$ is the common insertion point of three muscles of triceps brachii, and it is always on the red circle. The red circle represents the trochlear of humerus. $G$ is the starting point of brachial muscle, $H$ is the insertion point of brachial muscle, and $I$ is the position after rotation of $H$ point. $\theta$ is the elbow flexion angle. $F_{Bilong}$ and $F_{Bishort}$ represent the strength of long head and short head of biceps brachii respectively. $F_{Trlong}$, $F_{Trlat}$ and $F_{Trmed}$ represent the muscle strength of long head, lateral head and medial head of triceps brachii, respectively. $F_{Bra}$ represents the muscle strength of brachial muscle.

Based on the above anatomical model, combined with the classic Hill muscle model (Appendix A), the elbow muscle torque prediction process is shown in Figure 7 [36], and it is briefly described below. Firstly, the sEMG signals of six muscles were processed to get the muscle activation $a_1$. Then, according to the joint flexion angle $\theta$, the real-time muscle fiber length $l_m$, tendon length $l_t$ and muscle force arm $r_m$ were obtained using forward kinematic analysis. The cosine of the pinnate angle $\varphi$ can be obtained from $l_m$, and the normalized muscle fiber contraction velocity $v_n$ can be obtained by deriving $l_m$. By $l_m$, $\cos\varphi$ and $l_t$, muscle real-time length $l_{mt}$ can be obtained. Finally, according to Hill muscle model, the muscle fiber length influence factor $f_l$ can be obtained from $l_m$. According to $v_n$, the influencing factor of muscle fiber velocity $f_v$ is calculated. According to $a_1$, $f_l$ and $f_v$, the active coefficient $f_{CE}$ can be obtained, the damping coefficient $f_{VE}$ can be obtained from $v_n$, and the passive coefficient $f_{PE}$ can be obtained from $l_{mt}$. The sum of $f_{CE}$, $f_{VE}$ and $f_{PE}$ is calculated, and then the muscle strength $F_M$ is obtained by multiplying the cosine value

of feather angle $\cos\varphi$ and the maximum muscle strength $F_0$ of muscle fiber. The muscle strength $F_M$ is multiplied by $r_m$ to get the torque $\tau_1$ produced by the muscle in the joint. In this prediction model, the sEMG signals of muscles can be obtained by arranging muscle electrodes on the surface of related muscles, and the elbow rotation angle can be obtained by installing inertial sensors on the forearm. By analogy, we can calculate the torque of the other five muscles in the elbow joint. The resultant moment of elbow joint muscle $T_{muscle}$ will be obtained.

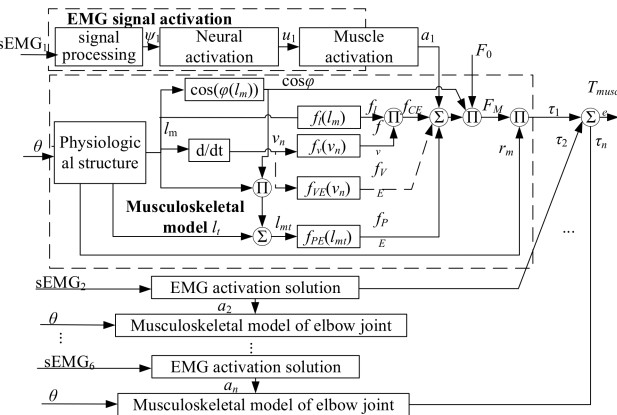

**Figure 7.** Prediction of elbow muscle torque based on musculoskeletal model.

Since the speed and acceleration of the upper limb are small when the patient wears the flexible exoskeleton, only the static force analysis is done. When the shoulder joint does not move, the load torque of elbow joint is the gravity torque of forearm, hand and end load, and these loads satisfy the following Equation (3),

$$T_G = M_l g l_{lcs} \sin\theta + (M_h + M_L)g(l_l + l_{hcs}) \sin\theta \tag{3}$$

In Equation (3), $M_l$, $M_h$ and $M_L$ are forearm mass, hand mass and hand load mass respectively; $G$ is the acceleration of gravity; $l_{lcs}$, $l_l$ and $l_{hcs}$ were the center of mass of forearm, forearm length and the center of mass of hand. According to $T_G$ and $T_{muscle}$, $T_{assist}$ can be calculated.

$$T_{assist} = T_G - T_{muscle} \tag{4}$$

### 3.2. Assist Torque Decomposition

Since the assist torque is finally provided by two cables, it is necessary to decompose the assist torque command $T_{assist}$ into the force command of two cable actuators. First of all, it is necessary to obtain the flexion arm $r_B$ and the extension arm $r_T$ of the two cable actuators, respectively. Figure 8 is the geometric analysis diagram of flexible exoskeleton. $P_1$ and $Q_1$ are the cable sheath anchor points and forearm cable anchor points, $P_2$ and $Q_2$ are the vertical points from the above anchor points to the limb axis, and $O$ point is the rotation center of elbow joint. It can be seen from Figure 8 that $l_5$ is the flexion assist force arm $R_b$ and $l_6$ is the extension arm $R_t$. Since the force arm in the extension direction of elbow joint is basically constant in physiological structure [37], the length of $l_6$ can be regarded as fixed, while $l_5$ needs to be solved.

According to the coordinate system of elbow joint musculoskeletal model set in Figure 6, when the elbow joint rotates to a certain angle, set the current coordinates of $P_1$ and $Q_1$ as $(x_{P1}, 0, z_{P1})$ and $(x_{Q1}, 0, z_{Q1})$ respectively, and the straight line $P_1Q_1$ satisfies Equation (5),

$$z - z_{P_1} = \left(\frac{z_{Q_1} - z_{P_1}}{x_{Q_1} - x_{P_1}}\right)(x - x_{P_1}) \tag{5}$$

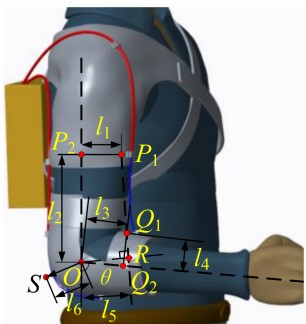

**Figure 8.** The geometric analysis diagram of flexible exoskeleton.

Thus, the flexion assist force arm $l_5$ is:

$$l_5 = \frac{\left| x_{Q_1} z_{P_1} - x_{P_1} z_{Q_1} \right|}{\sqrt{(z_{Q_1} - z_{P_1})^2 + (x_{P_1} - x_{Q_1})^2}} \tag{6}$$

When the elbow joint angle is 0, it can be seen from Figure 8 that $P_1$'s coordinate is $(l_1, 0, l_2)$, and $Q_{10}$'s ($Q_{10}$ is the initial position of $Q_1$ when the elbow is fully extended) coordinate is $(l_3, 0, -l_4)$. When the joint rotates, there is the following transformation formula,

$$\overrightarrow{OQ_1} = R\overrightarrow{OQ_{10}} \tag{7}$$

$$R = \begin{bmatrix} \cos\theta & 0 & -\sin\theta \\ 0 & 1 & 0 \\ \sin\theta & 0 & \cos\theta \end{bmatrix} \tag{8}$$

In Equations (7) and (8), $\overrightarrow{OQ_1}$ and $\overrightarrow{OQ_{10}}$ are the directed vectors from the origin $O$ to point $Q_1$ (when the flexion angle of elbow joint is $\theta$) and from the origin $O$ to point $Q_{10}$ (when the flexion angle of elbow joint is 0) respectively, and $R$ is the transformation matrix. According to Equation (7), the real-time coordinates of $Q_1$ is ($l_3 cos\ \theta + l_4 sin\ \theta$, 0, $l_3 sin\ \theta - l_4 cos\theta$). The flexion assist force arm $l_5$ can be obtained by substituting coordinates of $P_1$ and $Q_1$ into Equation (7), and it is shown in Equation (9),

$$l_5 = \frac{\left| (l_3 \cos\theta + l_4 \sin\theta) l_2 - (l_3 \sin\theta - l_4 \cos\theta) l_1 \right|}{\sqrt{(l_3 \sin\theta - l_4 \cos\theta - l_2)^2 + (l_1 - l_3 \cos\theta - l_4 \sin\theta)^2}} \tag{9}$$

According to the assist torque obtained from Equation (4) and the flexion assist force arm obtained from Equation (9), the flexion assist force on the cable actuator is,

$$F_B = \frac{T_{assist}}{l_5} \tag{10}$$

In Figure 9a, the exoskeleton flexion force arm is calculated according to Equation (9) when elbow joint flexing from 0 to 90 degrees. The initial coordinates of $P_1$ are (0.040, 0, 0.200), and the initial coordinates of $Q_1$ are (0.040, 0, −0.050). It can be seen from the figure that the cable flexion force arm first increases and then decreases. It peaked between 50°and 60°. In Figure 9b, the force on the flexion assist cable actuator can be obtained by using Equation (10). The force on the flexion assist cable increases with increasing of the elbow flexion angle. When the elbow flexion angle reaches 90°, the flexion cable force reaches the maximum, about 42 N.

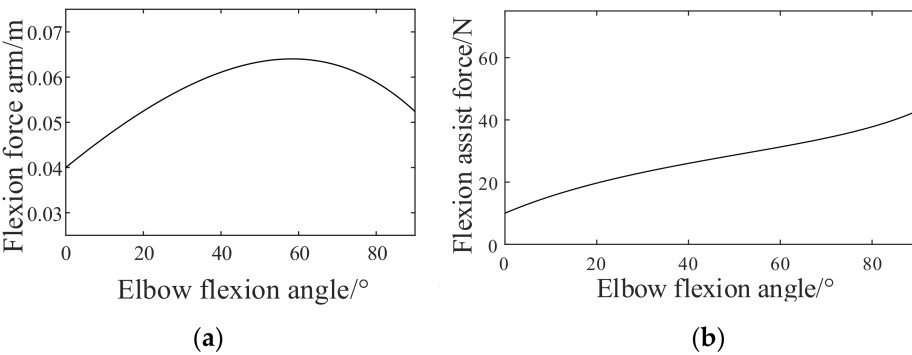

**Figure 9.** Force arm and force of flexion assist cable actuator. (**a**) Force arm. (**b**) Force on the flexion cable actuator.

According to the physiological model of elbow joint and the method of redundant muscle force analysis, the extension muscle only provides resistance and plays a stabilizing role when the elbow moves [38]. The force arm of the extension assist cable is set to be a constant 0.040 m for specific person. The stretching force moment is set to be 0.1 Nm, so the force on the extension assist cable actuator is 2.5 N.

*3.3. Force Control of the Nonlinear Cable Sea*

This section establishes the theoretical model of the nonlinear SEA mechanism, and then introduces the force control method. For the motor in the nonlinear cable SEA,

$$\tau_m = \frac{1}{k_{gear}} F_c r + J_m \frac{d^2 \theta_m}{dt^2} \tag{11}$$

In Equation (11), $\tau_m$ is the output torque of the motor, $k_{gear}$ is the reduction ratio of the planetary reducer, $r$ is the radius of the winding wheel, and $J_m$ is the moment of inertia of the motor, $\theta_m$ is the angular displacement of the motor. Assuming that the displacement of the nonlinear cable SEA is $l_L$ (when the end of the cable does not move, $l_L = 0$; when the end of the cable moves, $l_L$ can be calculated according to the coordinates of $P_1$ and $Q_1$ in Figure 8), the angular displacement $\theta_m$ of the motor is,

$$\theta_m = \frac{k_{gear}(l_L - g(\xi))}{r} \tag{12}$$

In Equation (12), $g(\xi)$ represents the change of cable length caused by nonlinear spring deformation, which can be calculated according to the geometric relationship of driving mechanism, and it satisfies Equation (13),

$$g(\xi) = \sqrt{a^2 + b^2} + \sqrt{a^2 + (b+c)^2} - \sqrt{a^2 + \xi^2} - \sqrt{a^2 + (\xi + c)^2} \tag{13}$$

Equation (1) gives the relationship between the force $F_c$ of cable SEA and $\xi$ when the friction force is ignored. However, in the working process of the cable driving mechanism, friction will inevitably occur between the cable, the pulley, the cable sheath, etc. In this paper, only the friction between the cable and the pulley is considered, and the friction between the cable and the cable sheath is not considered. The influencing factors of the friction between the cable and the pulley involve the roller material, the angle of the cable wrapping the pulley, the load of the cable actuator, etc., and the relevant literature has analyzed and modeled the influencing factors of this kind of cable drive mechanism in detail [39]. Previous studies have shown that the influence of cable angle on friction is very small. Secondly, the pulley in this paper is a rolling pulley with bearing. Therefore, the friction between the cable and the pulley mainly comes from the size of the load on

the cable. Let the friction coefficient related to the load be $k_1$, and it satisfies the following Equation (14),

$$k_1 = \frac{F_{c1} - F_c}{F_{c1}} \tag{14}$$

In Equation (14), $F_{c1}$ is the end force of the cable actuator after friction compensation. According to the existing research, the model of load and friction coefficient conforms to the characteristics of exponential function [39],

$$k_1 = u \cdot F_{c1}{}^v + w \tag{15}$$

In Equation (15), $u$, $v$ and $w$ are shape parameters, which have no practical significance and can be fitted by experimental data. After friction compensation, the force $F_{c1}$ at the end of the rope driver is,

$$F_{c1} = \frac{1}{(1 - k_1)} \frac{k(b - \xi)\sqrt{(a^2 + \xi^2)[a^2 + (\xi + c)^2]}}{\xi(\sqrt{a^2 + (\xi + c)^2} + \sqrt{a^2 + \xi^2}) + c \cdot \sqrt{a^2 + \xi^2}} \tag{16}$$

From Equations (11)–(13) and (16), the rotation angle $\theta_m$ of the motor can be obtained under the given force command $F_{c1}$. The common SEA controller often adopts the force control strategy with outer force loop and inner position loop. In this paper, the cascade controller with speed inner loop and force outer loop is adopted, and the motor in SEA is regarded as a speed source rather than a torque source, which can overcome the disturbance factors of viscous friction and mechanical gap in motor and reducer in force control, and it also ensure the passivity of the controller [40].

As shown in Figure 10a, the system block diagram of the force control of the nonlinear cable SEA in this paper is presented. According to the compression amount of the spring, the cable force $F_{c1}$ can be obtained according to Equation (16). After the error between the force command $F_{ref}$ ($F_B$ and $F_T$, which have been gotten before) and the end force feedback $F_{c1}$ passes through the PID controller, the motor speed command $\omega_{ref}$ is obtained. The motor output angular displacement $\theta_m$ is obtained by integrating the speed. After passing through the gearbox, the angular displacement of the winding wheel is $\theta_{m1}$, the cable displacement is $l_{motor}$. As shown in Figure 10b, the motor adopts double closed-loop control with speed outer loop and current inner loop.

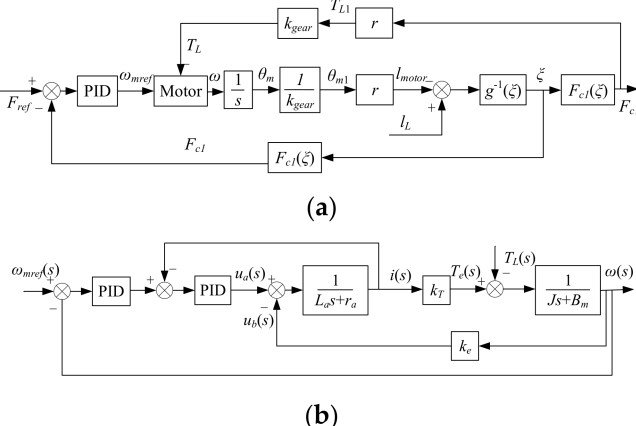

(a)

(b)

**Figure 10.** Tension control block diagram. (**a**) Cascade controller of the nonlinear cable SEA. (**b**) Motor control block diagram.

## 4. Experimental Verification

In order to verify the correctness of the control scheme, this section establishes an experimental platform to simulate the coupling between human and exoskeleton, and then

the force control of nonlinear cable SEA and the AAN control method of the flexible power assist exoskeleton are tested. Finally, the human wearing experiment is carried out.

*4.1. Construction of Experimentabl Platform*

The basic requirement of the experimental platform is that it should have similar transmission path with the exoskeleton. The load and force arm of the cable should be close to the actual working condition of the exoskeleton. Figure 11 is the schematic diagram of the experimental platform designed in this paper. The controller controls the two cable drivers on the flexion and extension sides, and the force sensor is added to the cable path of the cable actuator to observe the cable force. The cable finally acts on a rotating disk, and the rotating disk is coaxial with the rotating center of the load arm. The torque sensor is added to observe the load torque in the middle, and the angle sensor is added to observe the rotation angle of the rotating disk. The load arm is designed according to the size and weight of the forearm of the human body, and a heavy object can be hung at the end of the load arm as a load. After the experiment, the data of force sensor, torque sensor, and angle sensor can be imported into the computer for processing and analysis.

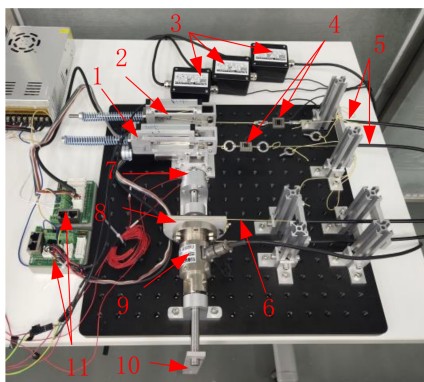

**Figure 11.** Schematic diagram of experimental platform. 1, extension side cable SEA; 2, flexion side cable SEA; 3, signal converter of tension sensor; 4, tension sensor; 5, cable sheath; 6, cable at drive end; 7, angle sensor; 8, rotating disk; 9, torque sensor; 10, load arm; 11, motor driver.

As shown in Figure 12, the two threaded holes on the rotating disk of the test bench are used to fix the cable in the flexion direction and the extension direction respectively (the red line is the cable). The cross section of the rotating disk can be regarded as a combination of rectangle and semicircle. The semicircular part was used to simulate the trochlear structure of humerus and ulna. The structure makes the extension assist cable move along the circular arc, and the extension assist force arm remains unchanged. Since the forearm has a certain volume, the rectangular part of the rotating disk is designed. When the elbow is flexing, the change of the flexion assist force arm is consistent with Figure 9a.

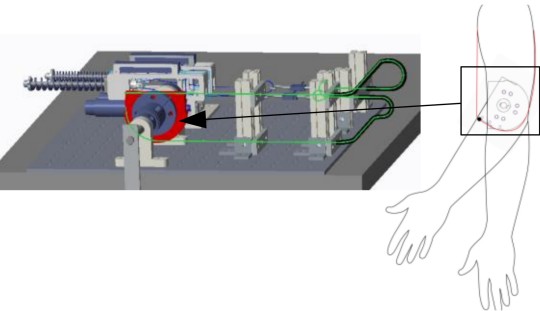

**Figure 12.** Design of rotating disk for simulating flexion and extension assist force arm.

### 4.2. Control Experimental Results

Firstly, the force control of the cable SEA is verified. The end of the nonlinear cable SEA is fixed, the sine force control command is given by the controller, and the force on the cable is collected by the force sensor. After fitting the test data, the shape parameter $u$ in Equation (15) is $-0.4077$, $v$ is $-0.603$, and $w$ is $0.247$. As shown in Figure 13a, it is the experimental result of sinusoidal force following of nonlinear cable SEA, in which the amplitude of input force command is 22 N and the period is 4 s. Figure 13b shows the tracking error of sinusoidal force command, in which the initial error is relatively large since the extension cable actuator is loaded to the preset force of 2.5 N at the beginning. Then the error changes periodically, and the peak value is within 1.5 N. It can be seen from Figure 13 that there is a good force following effect on the period of 4 s. Due to the slow movement of the elderly wearer, it can meet the requirements of the exoskeleton on the force control bandwidth.

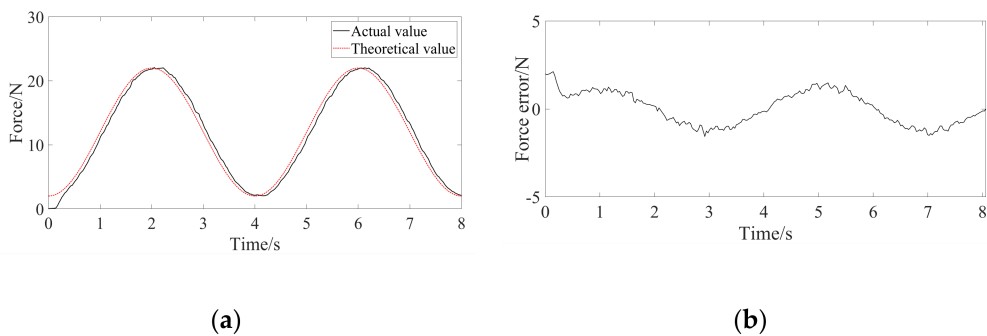

(**a**)  (**b**)

**Figure 13.** The sinusoidal force command following experimental results. (**a**) Sinusoidal command tracking curve. (**b**) Sinusoidal command tracking error curve.

Then, the AAN force control method is verified by experiments. In this paper, the sEMG signals of six muscles in musculoskeletal model shown in Section 3 were not actually collected. Instead, the open-source biomechanics simulation software OpenSim4.1 was used to obtain the sEMG signals under the specified motion state (elbow joint moved from $0°$ to $90°$ and then back to $0°$ within 4 s, and the hand load was 0 kg). After processing, the corresponding muscle activation was obtained as the input of elbow muscle torque prediction module (as shown in Figure 14).

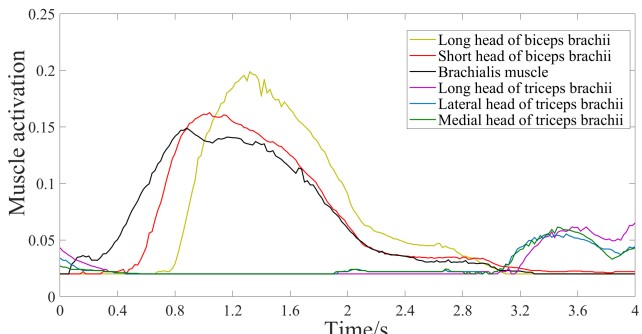

**Figure 14.** The related muscle activation obtained by OpenSim.

According to the method described in Section 3, the elbow muscle torque is predicted as shown in Figure 15. As can be seen from the figure, the maximum output muscle torque of the elbow joint is no more than 3 Nm. The solid line in Figure 15 represents the torque calculated by the musculoskeletal model in this paper, and the dotted line represents the torque calculated by OpenSim. After analysis, the Pearson correlation coefficient and $p$-value of the torque-time curve data under the two models are 1.390 and

0.239, respectively, indicating that the elbow musculoskeletal model established in this paper can predict the elbow muscle torque well.

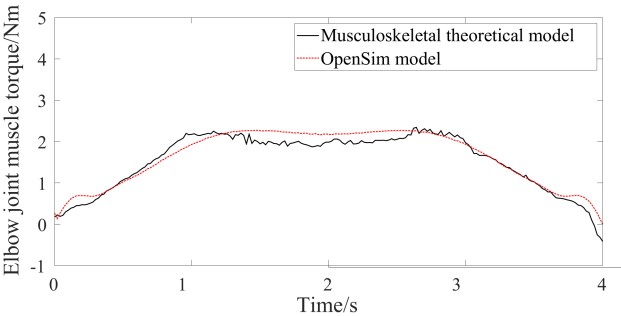

**Figure 15.** Predicted elbow muscle torque.

Assuming that the muscle torque output by the upper elbow joint of the assisted patient is only half of that of Figure 15, the assist torque provided by the flexible exoskeleton is 0.5 times the muscle torque predicted in Figure 15. According to the method in Section 3, the assist torque is decomposed to obtain the assist force command of single cable actuator, and finally the assist force of cable actuator is controlled. Figure 16 shows the response curve of the assist torque of the flexible exoskeleton. Compared with the two groups of curves, the torque error is limited between 0.06 Nm−0.02 Nm, and the actual assist torque well follows the predicted assist torque. Figure 17 shows the assist force command and actual assist force of the two cable actuators. The force of the cable on the flexion side and the extension side follows the force command well. The root mean square error of actual assist torque and command is 0.00226 Nm.

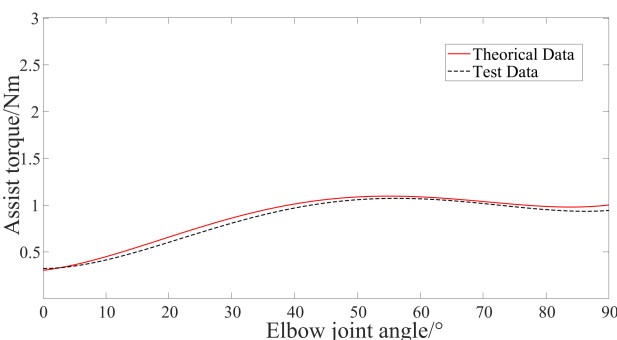

**Figure 16.** Exoskeleton assist torque control of elbow joint.

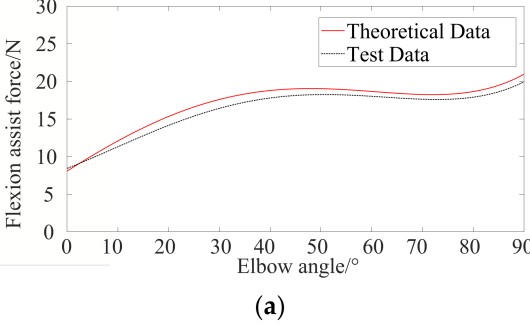

(**a**)

**Figure 17.** *Cont*.

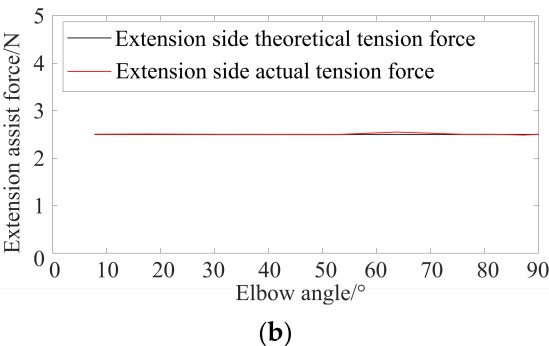

(b)

**Figure 17.** Assist force control experiment of antagonistic cable actuator. (**a**) Flexion side. (**b**) Extension side.

### 4.3. Preliminary Wearing Experiments

As shown in Figure 18, the wearing assist performance experiments are carried out finally. The height and weight of the wearer are roughly selected according to the data of the human model in OpenSim (height 1.7 m, weight 59 kg). The right arm of the wearer wears the flexible exoskeleton designed in this paper to flex and extend his elbow, and the surface EMG signals of the biceps brachii muscles of the upper arm are collected simultaneously. The wearer needs to do two groups of experiments. In one group, the wearer flexes and extends his elbow against the gravity of upper limb without exoskeleton assist torque. In the second group, according to the elbow torque predicted in Figure 15, 50% assist level is set, and the wearer complete elbow flexion and extension with the help of exoskeleton. Each group completes 10 exercise cycles. Since the cable actuator on the extension side is just for maintaining the tension of the elbow joint exoskeleton in extension direction, but not for providing assist force. It has little effect on the activation of triceps brachii, therefore, the EMG signal of triceps brachii was not collected.

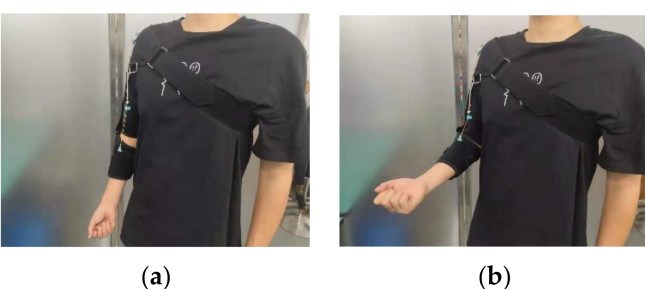

(**a**) 　　　　　　　　　　　　　　　　　　(**b**)

**Figure 18.** Flexible exoskeleton wearing experiment. (**a**) Extension state. (**b**) Flexion state.

Figure 19 is the activation degree of surface EMG signals of biceps brachii at 50% assistance level and without assistance. As can be seen from Figure 19, the activation degree of surface EMG signal of elbow flexor biceps brachii decreased significantly at 50% assistance level compared with without assistance, which preliminarily verifies the correctness of the AAN control method designed in this paper.

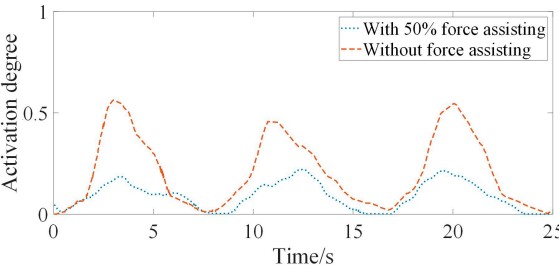

**Figure 19.** Comparison of activation of surface EMG signal of biceps brachii activation.

## 5. Discussion and Conclusions

It is an important trend to use flexible exoskeletons to assist patients with muscle weakness in daily life. In this study, a design scheme of a flexible upper limb exoskeleton with antagonistic cable actuators is proposed. An assist-as-needed force control method is studied, and preliminary experiments were carried out. Compared with the existing cable driven flexible exoskeleton, this paper uses the output force displacement characteristics of human upper limb muscle for reference in mechanical design. The cable is driven by a nonlinear series elastic actuator, and the elastic unit in the actuator simulates the nonlinear passive elasticity of human upper limb skeletal muscle. The muscle torque of the elbow joint is predicted by using the elbow joint musculoskeletal model with relatively complete muscle number and taking into account the humeral trochlear. Compared with OpenSim simulation data, Pearson correlation coefficient and *p*-value are 1.390 and 0.239, respectively, which shows that the model can predict muscle torque well. Then, based on the kinematic model of the flexible exoskeleton, the assist torque is decomposed into the force command of the cable actuators. The strategy of both a force outer loop and speed inner loop is used to control the cable actuator. Experiment results on the experimental platform show that the assist-as-needed force control method can realize the assist torque control well, and the root mean square error of actual assist torque and command is 0.00226 Nm. At the end of this paper, a preliminary wearing experiment is carried out to compare the activation of biceps brachii under different assist levels. The results verify the correctness of the method in this paper preliminarily. This study further developed the mechanical design and AAN force control method of the flexible upper limb exoskeleton, which has certain reference significance for the follow-up related research, but there are also some problems. First of all, the AAN control method proposed in this study needs to provide six muscle activation, upper limb related mass, geometric parameters and many other parameters, which is very cumbersome in practical application. It is also necessary to study how to reduce the required parameters on the basis of ensuring sufficient accuracy. Secondly, due to the characteristics of flexibility, it is very difficult to establish the real flexible exoskeleton model accurately. Thirdly, this study ignores the common hysteresis characteristics of cable driven mechanism, and further research is needed to solve the influence of hysteresis characteristics. Finally, the PID algorithm used in this study is relatively simple, and more advanced and interactive algorithms will be considered in the future. At present, some researchers use machine learning to model the kinematics of complex flexible cable driven robot, and have achieved good results [41] which can be applied to subsequent research.

**Author Contributions:** Conceptualization, B.H.; investigation, H.L. and H.Z.; methodology, H.L.; project administration, H.Y.; resources, H.Y.; software, F.Z.; supervision, B.H.; validation, J.Y. All authors have read and agreed to the published version of the manuscript.

**Funding:** This work was supported in part by Natural Science Foundation of Shanghai China (20ZR1437800).

**Institutional Review Board Statement:** Ethical review and approval were waived for this study, due to no enforced nor uncomfortable restriction to the human wearer during the study period.

**Informed Consent Statement:** Informed consent was obtained from all subjects involved in the study.

**Data Availability Statement:** Not applicable.

**Conflicts of Interest:** The authors declare no conflict of interest.

## Appendix A

The Hill muscle model consists of series elastic element (SEE), passive elastic element (PEE), contraction element (CE), viscous damping element (VE) and pinnate angle $\varphi$. In Figure A1, $l_{mt}$ is the muscle length, $l_m$ is the muscle fiber length, $l_{t1}$ and $l_{t2}$ are the tendon length at both ends.

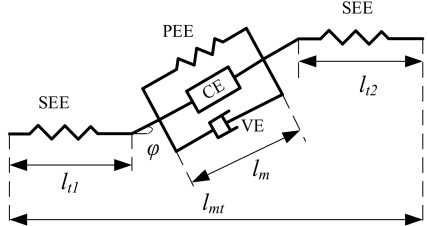

**Figure A1.** Hill muscle model.

The relationship among muscle length, muscle fiber length and tendon length are as follows:

$$l_{mt} = l_m \cdot \cos \varphi + l_t = l_m \cdot \cos \varphi + l_{t1} + l_{t2} \tag{A1}$$

The formula of muscle force is as follows:

$$F_M = (F_{CE} + F_{PE} + F_{VE}) \cdot \cos \varphi \tag{A2}$$

In Equation (A2), $F_{CE}$, $F_{PE}$ and $F_{VE}$ are the active forces of muscle fibers, the passive forces of muscle fibers, and viscous damping forces. The size of $F_{CE}$ is determined by muscle activation, muscle fiber length, muscle fiber contraction speed and maximum muscle force. The formula of $F_{CE}$ is as follows:

$$F_{CE} = a \cdot f_l \cdot f_v \cdot F_0 \tag{A3}$$

In Equation (A3), $a, f_l, f_v$ and $F_0$ are the muscle activation degree, the influence factor of muscle fiber length, the influence factor of muscle fiber contraction speed and the maximum isometric contraction force at rest, respectively.

In Equation (A3), muscle activation $a$ is calculated by the following formula:

$$a(t) = \frac{e^{Au(t)} - 1}{e^A - 1} \tag{A4}$$

In Equation (A4), $a(t)$, $u(t)$ and $A$ are muscle activation, normalized EMG signal and nonlinearity.

The influence factor $f_l$ of muscle fiber length and the influence factor $f_v$ of muscle fiber contraction speed in Equation (A3) were calculated by the method of Thelen model.

$$f_l = e^{-\frac{(l_m/l_{mopt} - 1)^2}{\gamma}} \tag{A5}$$

In Equation (A5), $l_m$ and $l_{mopt}$ are the real-time muscle fiber length and resting muscle fiber length. $\gamma$ is the shape coefficient, and 0.5 for the elderly and 0.45 for the young. The calculation formula of $f_v$ in Equation (A3) is:

$$fv = \begin{cases} \frac{1 + v_n}{1 - \frac{v_n}{A_s}}, & v_n \leq 0 \\ \frac{f_M \cdot v_n + \frac{A_s \cdot (f_M - 1)}{2 + 2 \cdot A_s}}{v_n + \frac{A_s \cdot (f_M - 1)}{2 + 2 \cdot A_s}}, & v_n > 0 \end{cases} \tag{A6}$$

In Equation (A6), $v_n$ is the normalized contraction velocity; $A_s$ is the curve parameter, and its value is 0.25; $f_M$ is the maximum force of muscle fiber elongation (normalizing the muscle fiber active force), which is 1.4 for young people and 1.8 for old people. The calculation of $F_{PE}$ can be obtained by multiplying $f_{PE}$ and $F_0$:

$$F_{PE} = f_{PE} \cdot F_0 \tag{A7}$$

As for $f_{PE}$, it can be calculated by the length of muscle fiber. When the length of muscle fiber is less than or equal to the rest length, no passive force will be generated. When the length of muscle fiber is greater than the rest length, passive force will be generated. Here we also use the formula of Thelen to calculate:

$$f_{PE} = \frac{e^{\frac{k^{PE}(l_m/l_{mopt}-1)}{\varepsilon_0^M}} - 1}{e^{k^{PE}} - 1} \tag{A8}$$

In Equation (A8), $k^{PE}$ is the curve shape parameter, and the value is 4; $\varepsilon_0^M$ was the maximum passive strain, 0.5 for the elderly. Adding viscous damping force $F_{VE}$ helps to simulate the ability of muscle to eliminate high frequency oscillation. However, the damping force of muscle at low speed is very small, so viscous damping force is ignored in this paper.

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
