# Peer review of "Design and Assist-as-Needed Control of Flexible Elbow Exoskeleton Actuated by Nonlinear Series Elastic Cable Driven Mechanism"

_actuators, doi:10.3390/act10110290_

Round 1

Reviewer 1 Report

This paper designs a flexible exoskeleton which is driven by two series elastic actuators antagonistically and presents an assist-as-need force control method based on the prediction of the exoskeleton’s assist torque. The paper is well written, and idea is interesting. A major concern is that there are too many useless figures and better to combine for better presentation. Some other comments are given to help improve the paper.

  1. As far as I know, there are many typical studies that are not included. Please keep references up to date;
  2. Although the musculoskeletal model with six muscles can better predict elbow torque, the model is too complex and may not be practical, which needs further explanation by the author;
  3. The citation format of references 17.18 should be [17, 18]. There are the same problems in the citation of references such as 19, 20 and 21;
  4. The font format of formula parameters in the body is inconsistent. Please check and correct the full text;
  5. Serial numbers (a) and (b) in some figures are missing;
  6. The formula number in the appendix shall be consistent with the picture;
  7. Suggest to highlight the significance of the study, and make it clear how it it important.

Reviewer 2 Report

This paper presents a study on the design scheme of an elbow exoskeleton driven by flexible antagonistic cable actuators. The wearing experimental results also show that the AAN control method designed in this paper can reduce the activation of biceps brachii effectively when the exoskeleton assist level increases. The innovation and contribution of this paper sounds good. However, some suggestions might be considered.

(1) The hysteresis characteristics of the cable driven mechanism should be considered in the model and control method.

(2) Some units of the data are forgotten to write the contents, i.e. ‘The root mean square error of actual assist torque and command is 0.00226.’

(3) The control method might be improved with more advanced algorithms better than PID, such that could improve the man-machine interaction performance.

Reviewer 3 Report

The paper presents a flexible elbow exo-skeleton actuated by nonlinear series elastic cable driven mechanism. Compared with other cable driven flexible exoskeletons, the equipment uses the output force displacement characteristics of human upper limb muscle for reference in mechanical design.

The article is interesting and written at a high scientific level. The presentation method is good and in accordance with generally accepted standards in that area.

Variants of such exoskeletons are known in literature. Hence the novelty of this work is quite limited. The constructive solution presented in the paper is a fairly simple one, its innovative character being questionable.

The paper layout is generally correct and clear. The manuscript has a sufficiently comprehensive and in-depth literature review. The methodology is well presented. The results confirm the conclusions in the manuscript.

There are several points the authors need to address before the paper can be considered for publication.

  • Please do not use acronyms that have not been defined previously. First write out the full expression and then introduce the acronym for further utilization in the paper (g. first write Assisted As Needed, then use AAN).
  • Citations of papers in the References section should comply with the template. For example: [4-6], [17,18], [19-21], [26,27]. [31-33].
  • Figures 4 and 9: please provide an improved quality: the resolution is not sufficient.
  • Lines 292 - 294: replace l5 by l6 and l5 by l6.
